# Reduction of the Adverse Impacts of Fungal Mycotoxin on Proximate Composition of Feed and Growth Performance in Broilers by Combined Adsorbents

**DOI:** 10.3390/toxins13060430

**Published:** 2021-06-21

**Authors:** Anthony Christian Mgbeahuruike, Toochukwu Eleazar Ejiofor, Michael Ushie Ashang, Chiamaka Ojiako, Christian C. Obasi, Chuka Ezema, Obianuju Okoroafor, Mulunda Mwanza, Magnus Karlsson, Kennedy F. Chah

**Affiliations:** 1Department of Veterinary Pathology and Microbiology, Faculty of Veterinary Medicine, University of Nigeria, Nsukka 410001, Nigeria; fabianchristiano1990@gmail.com (C.C.O.); kennedy.chah@unn.edu.ng (K.F.C.); 2Department of Agricultural Education, Faculty of Vocational and Technical Education, University of Nigeria, Nsukka 410001, Nigeria; Toochukwu.ejiofor@unn.edu.ng (T.E.E.); michaelashang@yahoo.com (M.U.A.); Chiamaka.ojiako@unn.edu.ng (C.O.); 3Department of Animal Health and Production, Faculty of Veterinary Medicine, University of Nigeria, Nsukka 410001, Nigeria; chuka.ezema@unn.edu.ng; 4Department of Medicine, Faculty of Veterinary Medicine, University of Nigeria, Nsukka 410001, Nigeria; Obianuju.okoroafor@unn.edu.ng; 5Department of Animal Health, Faculty of Natural and Agricultural Sciences, North-West University, Mafikeng Campus, Private Bag X 2046, Mmabatho 2735, South Africa; mulunda.mwanza@nwu.ac.za; 6Department of Forest Mycology and Plant Pathology, Swedish University of Agricultural Sciences, 75007 Uppsala, Sweden; Magnus.karlsson@slu.se

**Keywords:** adsorbents, broilers, fungi, proximate composition, nutrient digestibility, mycotoxins

## Abstract

Synergistic interaction of adsorbents in reducing the adverse impacts of mycotoxin on performance and proximate composition of broiler feeds was investigated. Fungal growth was induced by sprinkling water on the feed. *S. cerevisiae* + bentonite, kaolin + bentonite or *S. cerevisiea* + kaolin adsorbent combinations (1.5 g/kg feed) were added and the feeds were stored in black polythene bags. An untreated group was kept as a positive control while fresh uncontaminated feed was used as a negative control. Mycotoxins were extracted from the feeds and quantified using reverse phase HPLC. Proximate composition, nutrient digestibility of the feeds, feed intake and weight gain of the broilers were measured. Deoxynivalenol (DON) concentration in the contaminated/untreated feed was 347 µg/kg while aflatoxin B1 (AFB1) was 34 µg/kg. Addition of bentonite and kaolin in the contaminated feed reduced AFB1 and DON to significantly lower levels. Feed intake and weight gain were low in the broilers fed the contaminated feed. The carbohydrate level was significantly (*p* < 0.05) reduced from 62.31 to 40.10%, crude protein digestibility dropped from 80.67 to 49.03% in the fresh feed and contaminated feed respectively. Addition of the adsorbents (*S. cerevisiae* and bentonite) significantly (*p* < 0.05) improved these parameters.

## 1. Introduction

Poultry feed is compounded from different ingredients, primarily cereals (rice, wheat, barley, oats, rye, corn, sorghum and millet), milling by-products (brans, hulls, pollards) and oil cakes (palm kernel, soybean, sunflower, rapeseed, peanut, linseed, cottonseed). These components (especially corn and corn by-products) are highly susceptible to fungal contamination and thus susceptible to mycotoxin contamination [1]. Mycotoxins are fungal metabolites produced by fungi of different genera, for example certain *Aspergillus* species produces aflatoxins (AF) [2], while certain *Fusarium* species produce zeralenone (ZEN) [3,4]. Trichothecene mycotoxins, including deoxynivalenol (DON), are produced by several fungal genera such as *Trichothecium*, *Stachybotrys*, *Myrothecium*, *Cephalosporium*, *Trichoderma*, *Penicillium* and *Fusarium* species [3,4]. The chemical composition, ingredients and nutritional quality of poultry feeds influence the growth of mycotoxin producing fungi [1]. High moisture and crude fat were reported to increase the mycotoxin content of feed [5]. Liu et al. [6] observed that defatted grains (soybean, peanut, corn, wheat corn endosperm and corn germ) showed a significant decrease in mycotoxin concentration when compared with the full-fatted ones. On the other hand, when the same seeds were treated with corn oil, mycotoxin production by *A. flavus* increased [6]. Hence, the presence of lipids enhances growth of mycotoxin-producing fungi and, in turn, mycotoxin content in feed, especially aflatoxin B1 (AFB1).

Fungal growth on feed and subsequent release of mycotoxins in the feed have significant effect on the proximate (feed nutrient) compositions of feeds [7]. Hence, when mycotoxin-containing poultry feeds are consumed by broilers, the ability of the birds to effectively digest the feed nutrients and efficiently convert the feed into meat production can be significantly affected. Some mycotoxins have been shown to exert detrimental effects on the gastrointestinal tract by altering the normal intestinal functions such as barrier function and nutrient absorption while some mycotoxins also affect the histomorphology of the intestines, thereby affecting nutrient digestibility [8]. Mycotoxins have varying bio-availabilities, while some are more rapidly absorbed, others get further along the gastrointestinal tract before their absorption can take place [9]. Studies have shown that trichothecenes have a negative effect on the viability of the intestinal cells [10]. Mycotoxins have been demonstrated to have negative effects on nutrient digestive efficiency, utilization of nutrients and microbial cells [11]. Mycotoxins can influence the composition and fermentation products of the intestinal microbiota and this can affect the health and performance of birds [12].

Studies have shown that mycotoxins increase the permeability of the intestinal epithelial layer in numerous species, which can result in excessive/uncontrolled leakage of substances into the animal, as well as affecting intestinal cell viability [13]. Mycotoxins can also reduce cell proliferation, thus reducing the intestine’s ability to repair and replenish itself [13]. A decreased absorption of glucose was observed following T-2 toxin and DON intoxication in the gastrointestinal tract resulting from suppressed expression of the *SGLT1* glucose transporter gene [9,13]. Shortening of villi height was also observed in trichothecene-treated birds [14]. The changes of the villi were due to activation of the apoptotic pathway by the trichothecenes, which in turn leads to nutrient malabsorption [15]. Trichothecenes were found to further increase intestinal permeability by lowering the expression of some proteins of the tight junction [16]. Trichothecene-treated cells showed a significant decrease in the number of goblet cells that secrete mucin, a substance that is involved in gut barrier function [17,18,19]. In some studies, it was demonstrated that ZEN induced cell death of intestinal epithelial cells without altering the cell integrity as indicated by trans epithelial electrical resistance, affecting nutrient digestibility [20,21,22]. Some studies have reported a negative effect of the *Fusarium* mycotoxin fumonisin B1 (FB1) on intestinal cell lines [23]. Several studies have reported the adverse effects of mycotoxins on intestinal health and nutrient digestibility in broilers [23,24,25]. Furthermore, high concentration of AF in broiler feeds has been reported to repress feed intake [2]. Birds fed aflatoxin-contaminated diets have been shown to suffer from lower feed consumption rate, reduced weight gain, increased mortality, low egg production, impaired resistance to infections and induced pathological damage to the immune-related organs [2,26]. Birds fed aflatoxin-contaminated feeds were found to have reduced body weight due to poor feed conversion rate [27]. Taken together, it is clear that mycotoxin-contamination of poultry feed is an important animal health issue as well as causing huge economic losses to farmers [28].

Different approaches including chemical, physical and biological methods have been adopted in the decontamination of poultry feeds. The efficacy of several adsorbents such as *Saccharomyces cerevisiae*, bentonite and kaolin in poultry feed decontamination have been extensively studied. However, there is limited information on the ability of these adsorbents to reduce the adverse impacts of mold-contaminated feed on proximate compositions of feeds and nutrient digestibility in broilers. Certain bacteria, including *Lactobacillus* species, *Bifidobacterium* species as well as *Enterococcus faecium* have been used to detoxify mycotoxins in the gut and they prevent the absorption of such contaminants when ingested in the feed [29]. The yeast *S. cerevisiae* has been used as probiotic feed and as adsorbent in mycotoxin-contaminated feeds [30]. *S. cerevisiae* and certain bacteria act as a sequestering agent in the gastrointestinal tract of animals through the formation of mycotoxin-microorganism complex [31]. Bentonite clay has a structural composition that gives a large surface area and this has made it an excellent candidate for AF decontamination in poultry feeds [26]. Kaolin clay is formed through intensive weathering of rocks rich in feldspar (granite, arkose, certain types of orthogneisses and migmatites) and it has a very high adsorbent capability. Kaolin feed additive has the ability to improve feed conversion ratio of many livestock species due to their adsorption potential of many mycotoxins [32]. Our previous findings using single adsorbents (*S. cerevisiae*, kaolin and bentonite) showed that kaolin had a more reducing effect against the mold/mycotoxin induced mycotoxicosis in broilers, followed by bentonite and *S. cerevisiae* [26]. From our findings, DON was reduced from 634.5 to 251 μg/kg by kaolin adsorbent while total AF (AFB1, AFB2, AFG1 and AFG2) in the feed was reduced from 53.272 to 18.154 μg/kg [26]. The same trend was observed for bentonite and *S. cerevisiae* in mycotoxin reduction. Also, feed/feed ingredients are substrates for multi-mycotoxin occurrence, using one adsorbent to detoxify feed with several mycotoxins may not yield optimal results. Therefore, we thought that pairing the adsorbents may give a more protective effects as the synergistic interaction between two adsorbents is more likely to give a better effect than using a single adsorbent. Furthermore, no study has investigated the synergistic interaction of these adsorbents (*S. cerevisiae*, kaolin and bentonite) in reducing the adverse effect of mycotoxin-containing feed on nutrient digestibility in broilers. Additionally, extensive studies have been done on ochratoxin A (OTA) and other mycotoxins such as AFs, trichothecenes and DON in poultry feeds [10,11,17,26]. Not much is known about the level of contamination of the poultry feeds in the tropics by ochratoxin B (OTB).

## 2. Results

### 2.1. Mycotoxin Concentrations in the Feed

Five mycotoxins were screened, viz: CPA, AFB1, AFB2, DON and OTB. The fresh (unmoistened untreated) feed had minimal concentrations of the five analyzed mycotoxins while the contaminated but untreated feed had pronounced concentrations, with DON reaching the highest concentration (387 µg/kg) followed by AFB1 (34 µg/kg, Table 1). Addition of the adsorbents had a reducing effect on the detectable concentration of the toxins. The contaminated feed supplemented with bentonite and kaolin had a more pronounced effect on AFB1, AFB2 and DON reduction than the other two combinations of adsorbents. The concentrations of AFB1, AFB2, DON and OTB mycotoxins in the contaminated feed treated with SC/kaolin combination was lower than in feed treated with SC/bentonite combination (Table 1).

### 2.2. Proximate Composition of the Feed Samples

Table 2 presents data on the means and standard deviations of proximate compositions of poultry feeds treated with or without combined adsorbents. The carbohydrate (CHO) level in the fresh/unmoistened feed was 62.31% while the contaminated/untreated feed had 40.10% CHO concentration (Table 2). The moisture content of the feed increased from 8.09 in the fresh/unmoistened) to 36.03% in the contaminated/untreated feed. The percentage crude ash slightly increased in the positive control (contaminated/untreated feed) while the fat components of the feeds was lower in the positive control when compared with the negative control (fresh/moistened feed). The crude protein slightly increased from 11.70 to 12.06% in the contaminated but untreated feed.

In the adsorbents-treated groups C (contaminated feed treated with *S. cerevisiae* and bentonite) and D (contaminated feed treated with *S. cerevisiae* and kaolin), the moisture content was significantly (*p* < 0.05) higher in comparison with the positive control (contaminated but untreated feed) and the negative control (fresh feed). The crude ash increased significantly (*p* < 0.05) in group C but was significantly reduced in the other adsorbent treated groups (D and E-Table 2). The crude fat content was significantly (*p* < 0.05) lower in the adsorbent treated groups compared with both the positive and negative controls whereas the crude fiber content was high in all the adsorbent treated groups in comparison with the controls. Crude protein was significantly (*p* < 0.05) reduced by the addition of the adsorbents while the carbohydrate content of the feeds was also significantly (*p* < 0.05) decreased by the adsorbent treatments (Table 2).

### 2.3. Feed Consumption and Growth Rates

Overall, there was no significant difference in feed intake between the treatment groups (*p* = 0.209). However, the birds given the fresh and uncontaminated feed had high feed intake when compared to the other groups (Figure 1). The feed intake in the group fed contaminated and untreated feed was low.

An increase in weight was observed in the group that were fed fresh feed (Figure 2). The group that consumed the contaminated feed had very low growth rate. The adsorbents slightly improved the growth rate with the weight gain relatively higher in the group that ate the contaminated feed supplemented with *S. cerevisiea* and kaolin followed by the those given the contaminated feed supplemented with *S. cerevisiea* and bentonite and lastly those that ate the bentonite/kaolin supplemented feed (Figure 2).

### 2.4. Nutrient Digestibility

The digestibility of the CP varied from 49.03% to 80.67% (Table 3); the variation was significant (*p* < 0.05). The highest digestibility of CP was recorded in chickens fed uncontaminated and untreated feed, followed by those fed contaminated and kaolin-bentonite treated feeds. A similar trend was observed for digestibility of CHO and DM (Table 3). For crude fat, the digestibility was significantly increased (*p* < 0.05) by the addition of the adsorbents. The bentonite/kaolin combination significantly (*p* < 0.05) increased the digestibility of CHO more than the *S. cerevisiae*/kaolin or bentonite combinations. The digestibility of crude ash was increased significantly (*p* < 0.05) by the addition of *S. cerevisiae* and bentonite in the contaminated feed.

## 3. Discussion

The experimental feed was sprinkled with water to induce fungal growth and mycotoxin production, which was evident in the increase in mycotoxin (CPA, AFB1, AFB2, DON and OTB) concentrations identified in this study. The moisture content of feed is an important indicator of the quality of feed and a key to safe storage [33]. The sprinkled water and the prevailing hot climatic condition favored the growth of mycotoxin-producing fungi [26]. High temperature and moisture with poor aeration during storage predisposes feeds to fungal growth and mycotoxin accumulation [5]. The concentration of DON and AFB1 were quite high in the contaminated but untreated feed (387 µg/kg and 34 µg/kg, respectively) and this correlated with the observed changes in the proximate composition of the feeds and their nutrient digestibility. The mycotoxins were also correlated with slightly increased percentage crude ash in the contaminated but untreated feeds. However, the crude ash was high in the contaminated feed treated with *S. cerevisiea* and bentonite when compared with the fresh feed, the same trend was observed for crude fibre in the contaminated feed treated with *S. cerevisiea* and bentonite/kaolin. The reason for this increase was not known but it is possible that the *S. cerevisiea* as an adsorbent may have contributed to the increase. The fat components of the feeds were considerably reduced. Fat in poultry feed helps in the adsorption of fat-soluble vitamins and increases palatability of feed [34]. Ash as a component of feed describes the inorganic (mineral) content of feed. When the ash level of feed is low, birds become pre-disposed to diseases and poor egg shell formation. The carbohydrate level in the contaminated but untreated feed was significantly reduced from 62.31 to 40.10%. All these changes were probably due to the heavy growth of fungi which may have utilized the available nutrients in the feed. The carbohydrates provided the major source of substrate for the fungal growth and subsequent production of mycotoxin. Combined effects of the different adsorbents in all the treatment groups did not result to much positive change in the proximate composition of the feeds. However, the crude ash level was significantly increased in the contaminated feed treated with *S. cerevisiae* and bentonite. The feed intake and growth rate were considerably low due to the presence of mycotoxins in the feed. Some studies have shown that feeds contaminated with mycotoxins at concentrations close to the maximum permissible level may cause decrease in weight gain and feed consumption rate [35,36]. Other studies have shown that almost the same concentrations had no deleterious effects on birds’ performances [37]. Greater adverse effect of mycotoxins on growth were seen in younger broilers than in older ones [38], while greater reduction in feed intake and growth rate were observed in older birds than younger ones [39]. The authors suggested that the older birds may have had higher feed intake (more mycotoxins consumption) than the younger ones and consequently, higher feed conversion ratio (FCR), and higher adverse effects. [39] observed that the growth performance of Ross broilers was suppressed when the birds were fed DON-contaminated feed. However, these parameters were improved by addition of adsorbents, especially by the combined effects of bentonite and kaolin, and *S. cerevisiea* and bentonite. Kaolin and bentonite have been previously reported to have mycotoxin-reducing ability in feed contaminated with DON and AFs [26]. Although not much is known about their mycotoxin-lowering ability, both adsorbents are commonly used to detoxify poultry feeds because they lack primary toxicity. *S. cerevisiae* had been reported to have excellent mycotoxin-binding ability but its efficacy in mycotoxin reduction is not as high as that of bentonite and kaolin [26].

The digestibility of the different feed components; crude protein, crude fat, carbohydrate and dry matter were significantly reduced in the contaminated but untreated feeds. The significant decline in digestibility values for all the nutrients in the contaminated and untreated feed was probably due to the high level of DON and AFB1, although the other mycotoxins may have contributed in the reduction in digestibility. Feeding broilers with diets contaminated with DON adversely affect the morphology of the small intestines and this could in turn affect nutrient digestibility [28]. Furthermore, histopathological analysis of the ileum region of Ross broilers fed DON-contaminated diets revealed shorter villi and shallower crypt than the control birds [39]. Increased dietary fumonisin B1 was suggested to alter the normal digestive and nutrient absorptive functions of the epithetlial lining of the gastrointestinal tract of birds. Further studies have reported the adverse influences of dietary fumonisin on normal epithelial morphology [40]. Ochratoxin A was reported to induce a similar deleterious effect on intestinal barrier function, indicating a role of mycotoxins in non-specific gastrointestinal tract hypo function in animals [41]. Reduced crude protein digestibility was observed in ducks fed AF-contaminated diet [42]. Also reduced apparent nutrient and energy digestibility were recorded at occasional doses of AF consumption [43]. However, conflicting reports of *Fusarium* mycotoxins on digestibility at occasional doses have been documented. While [44] recorded reductions in protein digestibility, the same researchers, in a separate and unrelated study, reported increased protein digestibility and net protein utilization [45]. Increased jejunal and illeal viscosity were observed in broiler chicks fed wheat-based diets but the intestinal viscosity was found to be drastically decreased when the wheat diet was inoculated with *Fusarium* spp. [46]. However, digestibility of the different nutrients was improved by the addition of adsorbents in the feed. In the current study, the three adsorbent combinations had a significant impact on digestibility of the nutrients except *S. cerevisiae*/bentonite combination where the digestibility of carbohydrate was low. Also *S.cerevisiae*/kaolin and bentonite/kaolin had negative impact on crude ash digestibility. Mycotoxin binders have the primary function of adsorbing mycotoxins and preventing their absorption in the gastrointestinal tract [47]. The combination of two or more adsorbents to detoxify mycotoxin-contaminated poultry feeds could be more effective in reducing the effect of the toxins on nutrient digestibility because, most of the adsorbing agents appear to bind to only a limited group of mycotoxins while showing very little or no affinity to others [48,49]. ZEN and DON administered individually, and in combination, negatively affected mesophilic aerobic bacteria in the intestine and significantly affected nutrients digestibility, but their negative impact was prevented by the combined application of kaolinites and activated charcoal [50]. Fumonisin B1 was found to decrease cell viability and proliferation in a concentration-dependent manner, reducing nutrient digestibility in broilers, which was prevented by the addition of *S. cerevisiae* [23]. Bentonite was reported to reduce sphinganine accumulation in intestinal epithelial cells thereby blocking mycotoxin-induced apoptosis and growth inhibition [51]. Maintaining intestinal integrity is essential for the proper functioning of epithelial cells. The height of the duodenal villi of broiler chickens was influenced by the inclusion of kaolin in the feed. The highest averages of villus height were observed in chickens that were fed feed with kaolin compared to the control treatment (without kaolin). The absorptive capacity of the intestine was reported to be directly proportional to the size of the villi and poultry with higher villi may have a better absorption of nutrients [52]. In conclusion, the different combinations of adsorbents had varying effect on reducing the effect of mycotoxins on nutrient digestibility. The *S. cerevisiae*/bentonite combination appeared to show a better impact on reducing the adverse effects of the mycotoxins, although the digestibility of carbohydrates was low in the birds that were given feeds with this treatment.

## 4. Materials and Methods

### 4.1. Study Site

The experiment was conducted at the Agricultural Education Section of the Poultry Unit, Animal Science Farms, University of Nigeria, Nsukka, Enugu State, Nigeria. The location lies within the rainforest zone of South-Eastern Nigeria at longitude 6°51′24″ N, latitude 7°23′45″ E and altitude 396 m above sea level. The climate is humid with a mean annual rainfall of 1579 mm. The mean annual temperature, humidity, precipitation and wind are 24.9 °C, 85%, 73% and 13 km/h, respectively.

### 4.2. Feed Contamination

Twenty-eight bags (25 kg each) of Finisher Broiler feeds were purchased from a commercial wholesaler. The 28 bags of feed were poured on a clean cement floor and mixed thoroughly, before pouring back into different 25 kg bags. Each bag containing the feed (25 kg) was sprinkled with 1 L of water to get the feed wet enough to enhance the growth of naturally occurring molds. The moistened feeds were stored in sealed black polythene bags to keep heat and moisture content in order to promote fungal growth and subsequent production of mycotoxins. The feeds from all bags were mixed thoroughly at 2 days intervals for one week and kept for an additional five weeks to ensure an even distribution of mycotoxins. The contaminated feeds were then divided into four treatment categories: B (contaminated feed without adsorbent), C (treated with *S. cerevisiae* and bentonite at 1.5 g of each adsorbent/kg of feed), D (treated with *S. cerevisiae* and kaolin at 1.5 g of each adsorbent/kg of feed) and E (treated with bentonite and kaolin at 1.5 g of each adsorbent/kg of feed). The feed in each category was mixed thoroughly and kept in the dark nylon bag for another two weeks for the adsorbents to act properly. A fresh feed sample (from same source and batch) without moistening or addition of adsorbent was kept and used as a negative control (A). Ten 5 mL Eppendorf tubes (two for each feed treatment) were labeled accordingly and filled with different feed samples and stored at −20 °C until they were shipped for mycotoxins concentrations analyses. All the feed samples were analyzed for the determination of the dietary mycotoxins concentration before being administered to experimental birds. Ten additional Eppendorf tubes were labeled and filled with feed samples and stored at −20 °C for proximate and digestibility studies.

### 4.3. Experimental Design

Two hundred- and ten-day-old vaccinated Abor Acre broiler breed of chicks were sourced from a commercial hatchery firm and used for the study. The birds were randomly distributed into eight groups with three replicates of five birds each. The experimental birds were handled in accordance with the Federation of Animal Science Societies (FASS) guidelines for care and use of animals in research [53]. The chicks were subjected to standard brooding for four weeks in a deep litter brooding house with regulated charcoal-heated temperature. Routine vaccination was given to the birds during the brooding period. The birds were later moved to sub-pens measuring 1.5 m × 1.5 m and physically separated by wire gauze. Wood shavings were used as litter material. Normal prophylactic medication and vaccinations were administered according to the recommendation prescribed from the hatchery. Multivitamins were given at weekly intervals to boost the immune system of the birds and the birds were fed ad libitum. The birds were divided into five dietary groups using a completely randomized design and were placed on the different feed treatments (A- uncontaminated fresh feed, B- Contaminated and untreated feed, C- Contaminated and treated with *S. cerevisiae* and bentonite, D- Contaminated and treated with *S. cerevisiae* and kaolin, E- Contaminated feed but treated with bentonite and kaolin). The broilers used in this study were handled with strict compliance with the revised version of the Animals Scientific Procedures Act of 1986 for the care and use of animals for research purposes. Furthermore, approval to conduct the study was obtained from the institutional head of the Faculty of Veterinary Medicine, University of Nigeria Nsukka. The results from the study are reported with strict compliance with the procedures outlined in ARRIVE guidelines for reporting invivo experiments in animal research [54].

### 4.4. Mycotoxins Extraction

Five grams of each representative sample of feed (particle size between 0.5 and 1 mm) was weighed into a 50 mL polypropylene tube (Sarstedt, Nümbrecht, Germany) and 20 mL of the extraction solvent (acetonitrile/water/acetic acid 79:20:1, *v*/*v*/*v*) was added. Samples were extracted for 90 min at 524 rpm on a Labcon FS16 rotary shaker (Labcon, Chamdor, South Africa), after which it was centrifuged for 15 min at 3500 rpm. Sample extracts were diluted with the same volume of dilution solvent (acetonitrile/water/acetic acid 79:20:1, *v*/*v*/*v*), and the diluted extracts filtered through a NY 0.22 µm simple pure syringe nylon filter (Membrane Solutions, Tokyo, Japan) into vial bottles and injected into the chromatographic system. The samples were analyzed for the following mycotoxins: cyclopiazonic acid (CPA), aflatoxin B1 (AFB1), aflatoxin B2 (AFB2), DON and ochratoxin B (OTB).

### 4.5. Proximate Analysis of Samples

The proximate composition of raw poultry feeds such as moisture, crude protein, crude fiber, crude fat and total ash content were analyzed using the procedures described by [55]. For determination of moisture, two grams of poultry feed were weighed in a petri dish, placed in a hot air oven at 105–110 °C for a minimum of 6 h, and cooled in a desiccator. The process of heating and cooling was repeated until a constant weight was obtained. The percentage moisture was determined using the formula:(1)$$\%\;\mathrm{Moisture}=\frac{W2-\;W3}{W2-W1}\times 100\%$$
where W_1_ = Initial weight of empty dish, W_2_ = Weight of dish plus sample before drying and W_3_ = Final weight of dish plus sample after drying.

For Dry Matter (DM) determination, the weight of the empty dish was recorded and 2 g of sample was placed in the dish and the weight of both dish and sample was determined and recorded. The dish containing the sample was placed in an oven set at 100 °C for 24 h and later placed in a desiccator to cool and then weighed. The process of heating, cooling and weighing were repeated until a constant weight was obtained. The percentage dry matter was determined using the formula:% DM = 100 − % moisture content.(2)

Crude protein was determined using a Micro Kjeldahl distillation assembly procedure (Hexatec, Maharashatra, India). Two grams each of the samples were weighed and placed into a digestion tube and two Kjeldahl tablets added. Twenty mL of concentrated sulphuric acid (H_2_SO_4_) were added into the tubes and allowed to digest at 420 °C for 3 h. After cooling, 80 mL of distilled water was added into the digested solutions. Fifty mL of 40% caustic soda (NaOH) was added into 50 mL of digested solution and placed on the heating section of the distillation chamber. Thirty mL of 4% boric acid plus bromo cresol green and methyl red indicator was added into conical flasks of each sample and placed underneath the distillation chamber for the collection of ammonia. The solution was observed until it changed from orange color to green. About 0.1 mL of 1N HCl was measured into a burette. The conical flask that contained the solution was titrated until the color changed from green to pink. The burette reading was taken and the percentage crude protein computed using the formula:(3)$$\%\;\mathrm{CP}=\frac{\left(A-B\right)\times N\times 14.01\times 6.26\;\mathrm{mg}}{\mathrm{Weight}\;\mathrm{of}\;\mathrm{Sample}}$$
where A = mL of acid for titration; B = mL of acid for titrating blank sample; N = normality of acid used for titration.

For the determination of crude fat, two grams of each sample were weighed into a thimble and 200 mL of petroleum ether was measured into a conical flask. The crude fat (a combination of simple fat, fatty acid, esters, compound fat, neutral fat, sterols, waxes, vitamins A, D2, E, K, carotene and chlorophyll) soluble in ether was estimated by extracting in ether which was continuously volatilized at 60–80 °C, condensed and allowed to pass through the thimble containing the sample in a soxhlet apparatus (Borosil, Mumbai, India). The flask was removed, re-weighed and the percentage fat sample determined using the formula:(4)$$\mathrm{Crude}\;\mathrm{fat}\;\left(\%\right)=\frac{\mathrm{Weight}\;\mathrm{of}\;\mathrm{fat}}{\mathrm{Weight}\;\mathrm{of}\;\mathrm{sample}}\times 100$$

For crude fiber (including cellulose, hemicelluloses and lignin) determination, two grams of sample was successively digested with diluted acid (0.225 N) and alkali (0.313 N). Two grams of each feed sample was defatted with petroleum ether, the sample was boiled under reflux for 30 min in 200 mL of a solution containing 1.25 g of H_2_SO_4_ per 100 mL. The solution was filtered and the residue transferred into a beaker and boiled for 30 min with 200 mL of a solution containing 1.25 g of carbon-free NaOH per 100 mL. The final residue was then filtered and dried in an electric oven, weighed and incinerated. The incinerated sample was cooled, weighed and recorded. The loss in weight after incineration multiplied by 100 was the percentage crude fiber. Thus:(5)$$\mathrm{CF}\left(\%\right)=\frac{\mathrm{Difference}\;\mathrm{in}\;\mathrm{weight}}{\mathrm{Weight}\;\mathrm{of}\;\mathrm{sample}}\times 100$$

To find the total content of mineral matter of total ash i.e., the non-combustible portion of the feed, 2 g of sample were weighed in a silica crucible. The sample was ignited on a burner till smoke ceases. The crucible was placed on a muffle furnace and heated to 600 °C and kept for 2 h; at this point, a white or light gray ash was obtained. At this temperature, all organic matter was burnt leaving behind minerals. The crucible was removed from the furnace and cooled in a desiccator at room temperature and weighed again. Percentage ash was determined using the following formula:(6)$$\%\;\mathrm{Ash}\;\left(\mathrm{dry}\;\mathrm{basis}\right)=\frac{\mathrm{Weight}\;\mathrm{of}\;\mathrm{ash}}{\mathrm{Weight}\;\mathrm{of}\;\mathrm{original}\;\mathrm{sample}}\times 100$$

### 4.6. Feed Intake, Growth Rate and Nutrient Digestibility

The four birds in each treatment group were used for the growth rate measurements. Each bird was weighed on weekly bases for 5 weeks and the average growth rate per/bird in each group was recorded. The feed intake of birds in each treatment group was determined as the difference in the quantity of feed served and the quantity left after overnight feeding. The average daily feed intake per bird (ADFI) in each treatment group and the feed conversion ratios (FCR) were determined.

The total fecal collection method was used to determine nutrient digestibility. The experimental birds were enclosed in individual cages according to the respective dietary treatments described earlier. Feed and fecal samples were collected daily for 4 days using clean trays and prepared for determination of dry matter (DM) intake, crude protein (CP), crude fiber (CF), moisture, fat, carbohydrate and subsequent chemical analysis according to procedures in the 20th Edition of the Official Methods of Analysis of AOAC International [56]. For the dry matter analysis, 20 g each of the feed and fecal samples were dried at 65 °C and finely grounded with mortar and pestle to pass a 0.5 mm screen, and then stored in sealed containers for determination of DM, organic matter (OM), gross energy (GE), and amino acid (AA, except tryptophan). The samples were analyzed for DM by drying at 105 °C for 48 h. The dry samples were then burned in a muffle furnace for 3 h at 550 °C, and the ashes used to determine the content of crude ash. Organic matter was calculated by the difference between DM and crude ash. The GE was measured with an adiabatic bomb calorimeter (WHR-15 Oxygen bomb calorimeter, Biobase, Bioindustry (Chandong), Co., Ltd., Changsha, China). The apparent metabolisable energy and true metabolisable energy values of both samples were calculated by the method described in [57].

### 4.7. Data Analyses

The data generated from the experimental examination of feed intake, growth rate, nutrient digestibility and proximate compositions were subjected to Analysis of Variance (ANOVA) using Minitab Statistical Software version 18.1 (Minitab Inc., State College, PA, USA). Pairwise comparisons were performed using the Fisher’s least significant difference (LSD) method at the 95% significance level.

## Figures and Tables

**Figure 1 toxins-13-00430-f001:**
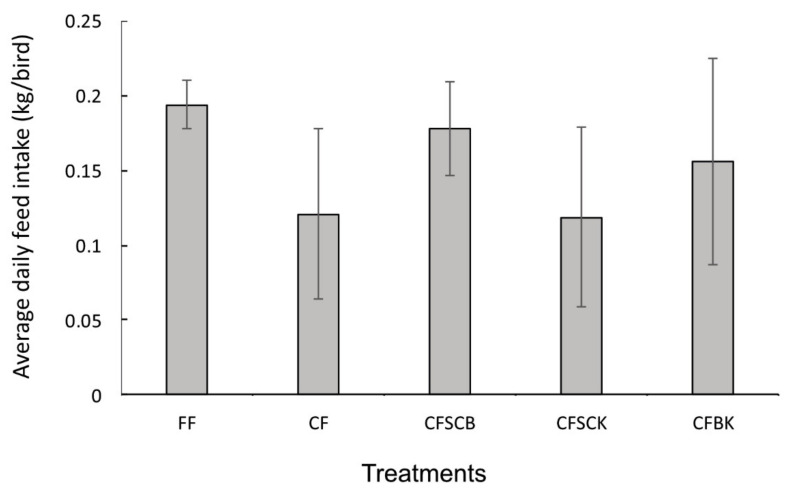
Feed intake measurements of broilers fed mycotoxin contaminated feed treated with or without combined adsorbents. Feed intake was measured during a period of five weeks, starting when the broilers were five weeks old. FF = fresh feed without mycotoxin contamination, CF = contaminated feed without adsorbent treatment, CFCSB = contaminated feed treated with *S. cerevisiea* and bentonite, CFSCK = contaminated feed treated with *S. cerevisiea* and kaolin and CFBK = contaminated feed treated with bentonite and kaolin. There were no significant differences in feed intake between treatment groups based on ANOVA analysis (*p* = 0.209).

**Figure 2 toxins-13-00430-f002:**
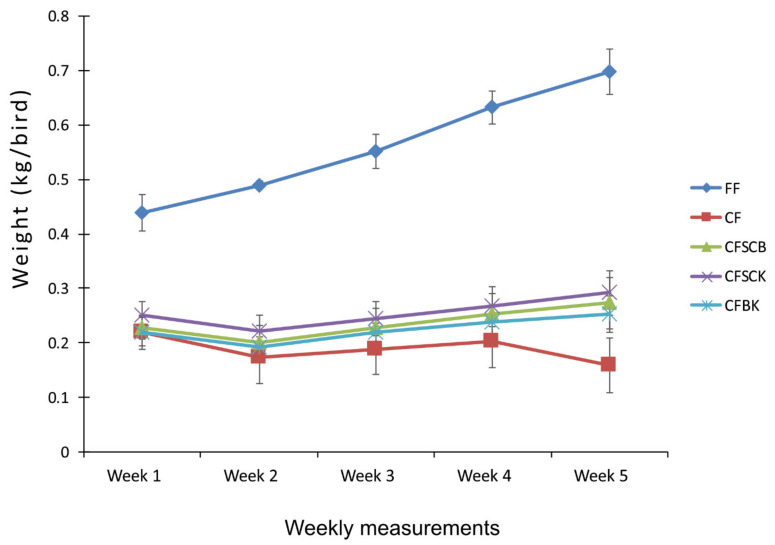
Weekly weight gain of broilers fed mycotoxin contaminated feed treated with or without combined adsorbents. Broiler weight was measured during a period of five weeks, starting when the broilers were five weeks old. FF = fresh feed without mycotoxin contamination, CF = contaminated feed without adsorbent treatment, CFCSB = contaminated feed treated with *S. cerevisiea* and bentonite, CFSCK = contaminated feed treated with *S. cerevisiea* and kaolin and CFBK = contaminated feed treated with benonite and kaolin. Weight gain in FF was significantly different (*p* ≤ 0.05) from the other treatments at all measurement time points based on the Fisher’s least significant difference test. Furthermore, CFSCB and CFSCK were significantly different from CF at the Week 5 measurement.

**Table 1 toxins-13-00430-t001:** Feed Mycotoxin Concentrations Following Different Adsorbent Treatments.

Feed Treatments	Mycotoxins (µg/kg)				
	CPA	AFB1	AFB2	DON	OTB
Fresh Feed	0.43	5.54	5.89	90.0	0.26
Contaminated + untreated	4.20	34.0	6.91	387.0	0.70
Contaminated + *S. cerevisiae* + Kaolin	0.59	15.4	2.57	286.0	0.41
Contaminated + *S. cerevisiae* + Bentonite	0.11	21.4	5.88	324.0	0.70
Contaminated + Bentonite + Kaolin	0.51	11.1	ND	92.0	0.47

CPA = Cyclopiazonic acid, AFB1 = Aflatoxin B1, AFB2 = Aflatoxin B2, DON = Deoxynivalenol, OTB = ochratoxin B. ND—not detected.

**Table 2 toxins-13-00430-t002:** Proximate Compositions of Poultry Feeds (%) Treated With or Without Adsorbents.

Group	Moisture	Crude Ash	Crude Fat	Crude Fiber	Crude Protein	Carbohydrates
A	8.09 ^d^	5.69 ^b^	5.78 ^a^	6.48 ^c^	11.7 ^a^	62.3 ^a^
B	36.0 ^c^	5.76 ^b^	0.17 ^d^	6.01 ^c^	12.1 ^a^	40.1 ^c^
C	43.0 ^b^	10.6 ^a^	3.31 ^b^	8.51 ^b^	9.55 ^b^	24.6 ^e^
D	45.5 ^a^	4.26 ^c^	2.52 ^bc^	10.4 ^a^	8.52 ^c^	28.6 ^d^
E	36.1 ^c^	4.39 ^c^	1.69 ^c^	6.56 ^c^	5.92 ^d^	46.3 ^b^

a = fresh feed, b = contaminated but untreated feed, c = contaminated feed but treated with *Saccharomyces cerevisiae* and bentonite, d = contaminated feed but treated with *Saccharomyces cerevisiae* and kaolin, e = contaminated feed but treated with bentonite and kaolin. Different letters within each variable indicate statistically significant differences (*p* ≤ 0.05) between treatments based on the Fisher’s least significant difference test.

**Table 3 toxins-13-00430-t003:** Nutrient Digestibility (%) of Broilers Fed Mycotoxin-Contaminated Feeds Supplemented With or Without Combined Adsorbents.

Group	Protein	Fat	CHO	Ash	Dry Matter
A	80.7 ^a^	69.1 ^a^	86.3 ^a^	36.3 ^c^	66.4 ^a^
B	49.0 ^d^	11.5 ^e^	69.9 ^b^	43.2 ^b^	46.9 ^c^
C	43.2 ^e^	65.2 ^b^	41.3 ^d^	73.8 ^a^	48.7 ^c^
D	53.8 ^c^	30.0 ^d^	60.7 ^c^	10.2 ^e^	40.7 ^d^
E	59.2 ^b^	57.6 ^c^	84.0 ^a^	13.8 ^d^	55.4 ^b^

CHO = Carbohydrate, a = fresh feed, b = contaminated but untreated feed, c = contaminated feed but treated with *Saccharomyces cerevisiae* and bentonite, d = contaminated feed but treated with *Saccharomyces cerevisiae* and kaolin, e = contaminated feed but treated with bentonite and kaolin. Different letters within each variable indicate statistically significant differences (*p* ≤ 0.05) between treatments based on the Fisher’s least significant difference test.

## Data Availability

Not applicable.

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
