# Peer review of "Reduction of the Adverse Impacts of Fungal Mycotoxin on Proximate Composition of Feed and Growth Performance in Broilers by Combined Adsorbents"

_toxins, 2021, doi:10.3390/toxins13060430_

Round 1
Reviewer 1 Report
Dear Authors,
The article titled: Reduction Of The Adverse Impacts Of Fungal Mycotoxin On Proximate Composition Of Feed And Growth Performance In Broilers By Combined Adsorbents is an interesting study where the synergistic interaction of adsorbents in reducing the adverse impacts of mycotoxin on performance and proximate composition of broiler feeds was investigated. In everyday life, there are more common problems with mycotoxins in feeds with normal moisture and not moldy feed as in this study. The weight gain, etc. are not only a result of mycotoxin action, but also of organoleptic changes in the feed. In addition, I need further explanation to understand your work:
- In my opinion, the most important explanation is missing, and that is - why did you choose to study the combination of feed additives and not the effects of each additive with the effects of different combinations. Please explain at the end of the introduction. The rationale that no one is studying the synergistic interaction of these absorbents is not sufficient.
2.Do you think that absorbents like bentonite or kaolin can reduce the effect of SC?
- How can you really prove that the combination is better than e.g. SC alone?
- Why did you decide to screen OTB? What about OTA?
- Table 2: further explanation is needed why the % crude fiber (group D) and ash (group C) is higher than group A in some cases.
- Line 294: The sampling method is very important as you need to achieve representativeness of the sample and further in the laboratory homogeneity of the sample. Did you collect the samples from different parts with special equipment, mix them and then prepare the final sample? Did you follow a protocol for proper sampling of feed for mycotoxins? Please describe this in more detail.
- Line 333: Proximate analysis of samples. The chapter is too long. You do not need to describe this basic analysis in detail with formulas. A proper reference is sufficient.
Author Response
Dear Editor,
Thanks for prompt review of our manuscript. We are revised the manuscript and the quality has improved tremendously. We have equally answered the reviewer’s queries one after as suggested.
Reviewer 2 Report
This is a high quality manuscript. I would only suggest that the authors edit data in terms of significant figures - data should be presented with up to 3 significant digits. There is no need for more digits (4 or 5).
Author Response
Dear Editor,
Thanks for prompt review of our manuscript. The suggested changes have been made and the quality of the manuscript has improved.